# Head-to-tail interactions of the coiled-coil domains regulate ClpB activity and cooperation with Hsp70 in protein disaggregation

**Marta Carroni[1], Eva Kummer[2], Yuki Oguchi[2], Petra Wendler[3], Daniel K Clare[1], Irmgard Sinning[4], Jürgen Kopp[4], Axel Mogk[2], Bernd Bukau[2], Helen R Saibil[1]***

[1]Department of Crystallography, Birkbeck College, University of London, London, United Kingdom; [2]Zentrum für Molekulare Biologie, Universität Heidelberg, Heidelberg, Germany; [3]Gene Center, Ludwig-Maximilians-University Munich, Munich, Germany; [4]Biochemie-Zentrum, Universität Heidelberg, Heidelberg, Germany

**Abstract** The hexameric AAA+ chaperone ClpB reactivates aggregated proteins in cooperation with the Hsp70 system. Essential for disaggregation, the ClpB middle domain (MD) is a coiled-coil propeller that binds Hsp70. Although the ClpB subunit structure is known, positioning of the MD in the hexamer and its mechanism of action are unclear. We obtained electron microscopy (EM) structures of the BAP variant of ClpB that binds the protease ClpP, clearly revealing MD density on the surface of the ClpB ring. Mutant analysis and asymmetric reconstructions show that MDs adopt diverse positions in a single ClpB hexamer. Adjacent, horizontally oriented MDs form head-to-tail contacts and repress ClpB activity by preventing Hsp70 interaction. Tilting of the MD breaks this contact, allowing Hsp70 binding, and releasing the contact in adjacent subunits. Our data suggest a wavelike activation of ClpB subunits around the ring.

**\*For correspondence:** h.saibil@mail.cryst.bbk.ac.uk

**Competing interests:** The authors declare that no competing interests exist.

**Reviewing editor**: Andreas Martin, University of California, Berkeley, United States

## Introduction

Cellular machinery has evolved to prevent or reverse protein misfolding and aggregation, which are damaging to cells and tissues. Bacterial ClpB and its yeast counterpart Hsp104 are members of the Clp/Hsp100 family and function in disaggregation and refolding of protein aggregates together with Hsp70 and its co-chaperones (*Glover and Lindquist, 1998*; *Goloubinoff et al., 1999*; *Motohashi et al., 1999*; *Zolkiewski, 1999*; *Doyle and Wickner, 2009*; *Haslberger et al., 2010*). ClpB/Hsp104 are conserved in bacteria, fungi, plants and mitochondria and are essential for recovery of cells from heat shock and other proteotoxic stresses (*Parsell et al., 1994*; *Mogk et al., 1999*). The oligomeric ring Hsp100 proteins thread substrates through a central channel, via binding to conserved tyrosine residues on flexible loops (*Weber-Ban et al., 1999*; *Kim et al., 2000*; *Lum et al., 2004*; *Weibezahn et al., 2004*; *Haslberger et al., 2008*; *Tessarz et al., 2008*). They belong to the AAA+ (ATPases associated with various cellular activities) superfamily of ATPases, with characteristic α and β subdomains (*Ogura and Wilkinson, 2001*; *Erzberger and Berger, 2006*). ATP binds between the two subdomains and at the subunit interface of adjacent monomers, with a catalytic Arg-finger provided by the neighbouring subunit.

While some other members of the Hsp100 family have been crystallised in their oligomeric form (*Bochtler et al., 2000*; *Glynn et al., 2009*; *Wang et al., 2011*), the atomic structure of ClpB is known only for the monomer (*T. thermophilus*). ClpB is composed of a N-terminal domain (ND) followed by two AAA+ domains (AAA-1 and AAA-2). An 85 Å long coil-coiled propeller, the middle domain (MD),

**eLife digest** Proteins are long chain-like molecules that twist and fold into complex three-dimensional shapes in order to carry out their functions. High temperatures or other types of stress can cause proteins to fold incorrectly, and misfolded proteins can form clumps (or aggregates) that are harmful to cells. Additional proteins called chaperones are therefore used by cells to help proteins to fold correctly, or to refold poorly folded proteins.

ClpB proteins (and related proteins) are chaperones found in bacteria, fungi and plants; these proteins co-operate with other chaperones to rescue misfolded proteins that have aggregated—an activity that helps cells to survive heat shock and other stresses. Six ClpB proteins work together to form a ring-shaped complex, and the misfolded protein is unfolded by threading it through the centre of this ring. Each ClpB protein also has a middle domain that acts to switch the complex on and off as needed.

The middle domains are known to form coiled-coils, with protein helices coiled together like the strands of a rope. However, previous efforts to work out the structure of the ClpB complex did not clearly establish where these coiled-coils were positioned relative to the rest of the ring.

Now Carroni et al. have used image processing to overcome these problems and reveal that the middle domains are wrapped around the outer edge of the ring complex. Analysis of ClpB mutants that lock the complex in either an off or on state revealed that the middle domains are linked head-to tail to encircle the ring when the complex is off. However, when the complex switches on, the middle domains let go of each other and tilt, allowing the ring to change shape. Carroni et al. suggest that the exposed ends of the middle domains are free to bind to other chaperones (those that work to refold the unfolded proteins), thereby activating the complex.

Although Carroni et al. have revealed how the ClpB ring complex is activated, further work is needed to understand exactly how the unlocked ring works to rescue misfolded proteins from aggregates within cells.

is inserted into the small subdomain of AAA-1 (*Lee et al., 2003*). It has two blades with mutationally sensitive sites at either end, termed motif 1 and motif 2. Cryo-EM reconstructions of *Tth*ClpB hexamers show a two-tiered molecule accounting for the AAA+ rings, but lack density for the ND. On the other hand, various cryo-EM studies of Hsp104 revealed the presence of an ND ring on top of the AAA+ ones. Although the overall shape and dimensions of the ClpB and Hsp104 hexamers are comparable in the various cryo-EM studies, there are substantial differences in the channel width (*Lee et al., 2003*; *Wendler et al., 2007*, *2009*; *Lee et al., 2010*). Observation of narrow (*Lee et al., 2003*, *2010*) (15–30 Å) vs expanded (*Wendler et al., 2007*, *2009*) (30–80 Å) cavities has led to different pseudo-atomic models of the hexamers. In one model, the AAA+ rings are compact, as in the crystal structures of other AAA+ hexamers such as p97, HslU or SV40 LTag, with the Arg-finger contacting the neighbouring subunit (*Bochtler et al., 2000*; *Huyton et al., 2003*; *Gai et al., 2004*). In the EM maps used to build this compact model there is little or no density for the MD and it was assumed to extend radially outwards from the ring (*Lee et al., 2003*, *2010*). In the expanded model, the AAA+ domains are more widely separated with the MD intercalated between them, preventing the canonical Arg-finger contacts (*Wendler et al., 2007*, *2009*). An attempt to localize the Hsp104 MD by genetically inserting a lysozyme resulted in a cryo-EM reconstruction of this chimera with visible density for the lysozyme, but not for the MD itself (*Lee et al., 2010*). Since none of the existing cryo-EM structures allows an unambiguous localization of the MD, its position is still a matter of debate (*Desantis and Shorter, 2012*).

The MD confers unique disaggregase ability to ClpB/Hsp104 (*Kedzierska et al., 2003*; *Mogk et al., 2003*) and is required for species-specific cooperation with the DnaK-DnaJ(DnaKJ)/Hsp70-Hsp40 system (*Sielaff and Tsai, 2010*; *Miot et al., 2011*; *Seyffer et al., 2012*). A direct interaction between the ClpB MD and DnaK has been shown by NMR spectroscopy and site-specific crosslinking and involves the motif 2 tip of the MD and the ATPase domain of DnaK (*Seyffer et al., 2012*; *Rosenzweig et al., 2013*). The MD acts to repress ClpB disaggregase activity, and DnaK binding relieves this repression (*Oguchi et al., 2012*; *Seyffer et al., 2012*). Point mutations in the MD show that interactions between motif 2 and AAA-1 are critical for regulating ATPase and disaggregase activities (*Haslberger*

*et al., 2007*; *Oguchi et al., 2012*; *Seyffer et al., 2012*; *Lipinska et al., 2013*). Thus, the MD plays an essential role in coupling Hsp70 interaction to ATPase regulation and substrate disaggregation in ClpB/Hsp104. It is therefore important to understand its structural role in ClpB/Hsp104 hexamers and in the context of the ClpB-DnaKJ bi-chaperone machinery.

To address the ambiguities in domain arrangement and to elucidate the working principle of the MD, we performed single particle EM studies of ClpB under conditions where its orientation could be determined more reliably than in previous studies. We used BAP (ClpB with the ClpA tripeptide for ClpP binding), a chimera engineered to bind the ClpP protease via the replacement of a C-terminal ClpB segment with the ClpP binding region of ClpA (*Weibezahn et al., 2004*). This construct has been extensively used to study the ClpB disaggregation mechanism by monitoring substrate proteolysis after delivery to ClpP (*Weibezahn et al., 2004*; *Haslberger et al., 2008*; *Tessarz et al., 2008*; *Mizuno et al., 2012*; *Rosenzweig et al., 2013*; *Figure 1A*). Therefore, BAP is suitable for structural studies and has allowed us to obtain maps with visible MD densities that shed light on its regulatory mechanisms.

## Results

### Three-dimensional (3D) reconstruction of negatively stained BAP-ClpP shows clear density for the middle domain

Using the BAP complex facilitates orientation determination in EM reconstruction, as previously observed for ClpA-ClpP (*Effantin et al., 2010*). Side views of the elongated BAP-ClpP complex are easily recognisable (*Figure 1B*) whereas in 2D projections of ClpB alone, which has a globular shape, it is hard to distinguish side from tilted views. By restricting the dataset to side views the angle assignment is more reliable, and these views are sufficient to generate the full 3D structure (*Figure 1C*). Using H/D exchange experiments, which report on the solvent accessibility and structural flexibility of amide hydrogens, we found that BAP, either alone or bound to ClpP, displays the same protection pattern as ClpB, implying the same MD conformation (*Figure 1—figure supplement 1*).

The BAP-ClpP complexes formed by mixing BAP and ClpP at a 1:1 molar ratio of hexamer to heptamer in the presence of ATPγS were stained with uranyl acetate. Complexes containing two BAP hexamers per ClpP double-heptamer were picked for single particle analysis and separated into halves for processing (*Effantin et al., 2010*; *Figure 1B*). Initial analysis revealed the presence of four layers corresponding to the ND, AAA-1, AAA-2 and ClpP rings. Class averages and eigenimage analysis indicate that the ND is extremely mobile in the BAP hexamer (*Figure 1—figure supplement 2*), consistent with crystallographic data on monomeric *T. thermophilus* ClpB (*Lee et al., 2003*). The ND layer was therefore excluded during later stages of image alignment. Similarly, the region corresponding to ClpP, very useful for the initial analysis, was not included in the refinement because of its symmetry mismatch with ClpB. Using ~12,000 particles, we obtained a 3D map by refining the alignment of the two AAA+ rings only, but then including the whole molecule in the reconstruction, which was at ~17 Å resolution (*Figure 1C,D*, *Figure 1—figure supplement 3*). In order to simplify the problems of alignment and reconstruction, we initially imposed sixfold symmetry, which blurs the features of the heptameric ClpP ring. Similarly, the mobile ClpB ND is blurred into a solid disc.

The BAP hexamer has overall outer dimensions of ~150 × 100 Å, similar to previous structures of ClpB/Hsp104 (*Parsell et al., 1994*; *Lee et al., 2003*; *Wendler et al., 2007*, *2009*; *Figure 1C*). It encloses a ~30 Å wide central channel, comparable in size to that in the crystal structure of ClpC (*Wang et al., 2011*; *Figure 1C,D*). In the reconstruction it is possible to identify regions accounting for all the domains, such as L-shaped densities for the AAA+ domains and a rod-like density for the coiled-coil MD.

To interpret the domain interactions, we fitted ClpB atomic coordinates into the EM density. We determined the crystal structure of an ND truncation of *E. coli* ClpB (residues 159 to 858; E279A/E432A/E678A mutation; *Table 1*). The subunit structure is very similar to that of *T. thermophilus* ClpB (*Figure 1—figure supplement 4*). Since none of the available crystal structure conformations fit in the EM map, domains were fitted as separate rigid bodies connected at hinge regions (*Figure 1C*). For the AAA-1 ring, it was possible to build a hexamer model based on the crystal structure of hexameric ClpC (*Wang et al., 2011*) (PDB code 3PXG), a homologue that also displays disaggregation activity in vitro (*Schlothauer et al., 2003*). The resulting ClpB AAA-1 hexamer model was automatically docked into the AAA-1 layer as a rigid body (*Figure 1D*). This strategy was chosen over the fitting of a single subunit followed by hexamerisation because it is expected to provide a

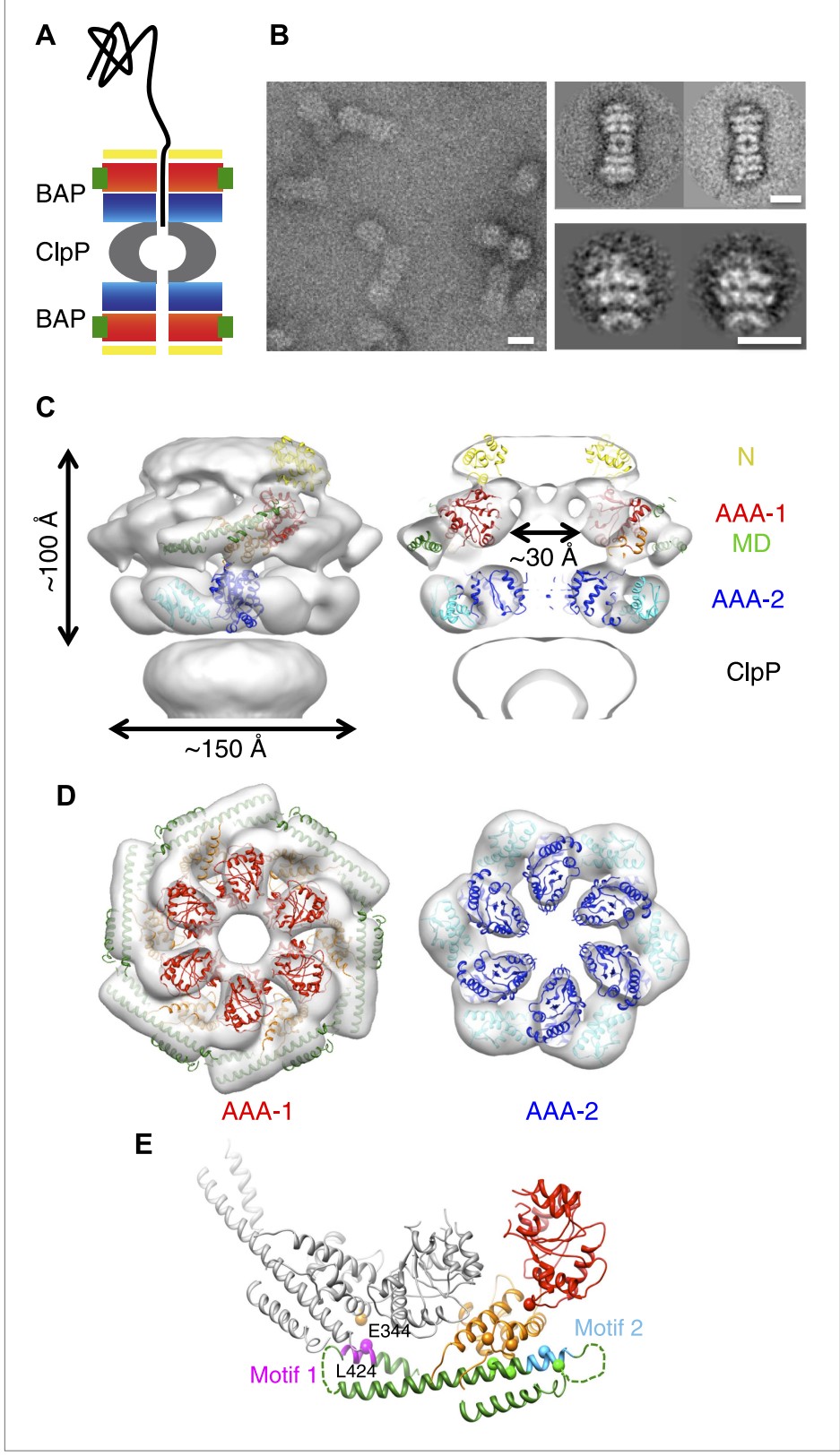

**Figure 1**. 3D structure of the symmetrised BAP-ClpP complex. (**A**) Schematic of the complex showing the threading of a polypeptide. ClpB domains are in yellow, N-terminus; red/orange, AAA-1 domain; green, MD and blue/cyan, AAA-2 domain. ClpP is in grey. (**B**) Negative stain EM image (left panel), class averages of 2:2 BAP-ClpP (upper

*Figure 1. Continued on next page*

*Figure 1. Continued*

right panel) and cut out 1:1 BAP-ClpP particles (bottom right panel). Scale bars are 150 Å. (**C**) BAP-ClpP structure with a fitted ClpB monomer (left) and central slice with fitted atomic coordinates (right). (**D**) AAA-1 and AAA-2 layers. (**E**) Two adjacent AAA-1 domains from the hexameric fit viewed in the plane of the ring. One monomer is in colour and the other in grey. Residues involved in engineered disulphide bonds are shown as spheres. Residues involved in intermolecular cross-links are labeled. The separation between cross-linking Cα pairs ranges from 8 to 14 Å.

The following figure supplements are available for figure 1:

**Figure supplement 1**. Complex formation between BAP and ClpP does not change MD protection pattern in HX experiments.

**Figure supplement 2**. The BAP N-terminus is highly mobile.

**Figure supplement 3**. Fourier Shell Correlation plots to estimate EM map resolution.

**Figure supplement 4**. Crystal structure of *E. coli* ClpB and EM based model.

more accurate picture of the subunit interface, which is difficult to determine at the resolution of our EM map. However, this approach did not work for the AAA-2 layer since the ClpB AAA-2 hexamer based on the ClpC crystal structure was not compatible with the density. In this case a ~40° tilt of the monomers into the plane of the ring was required to obtain the optimal fit, which resembles the pseudo-atomic model of the homologue ClpA AAA-2 ring (*Effantin et al., 2010*). Coordinates of the *E. coli* ClpB ND (residues 1 to 148) were obtained from the PDB (1KHY). A single N domain was fitted manually, maintaining the connection to AAA-1, and then hexamerised by applying symmetry in Chimera (*Pettersen et al., 2004*).

The AAA+ rings have a central opening of ~30 Å and therefore are not as compact as in one of the previous models (*Lee et al., 2003*, *2007*, *2010*), but not as expanded as in the other (*Wendler et al., 2007*). The Arg-fingers are at the interface between subunits, available to catalyse hydrolysis as expected from mutational studies (*Mogk et al., 2003*; *Yamasaki et al., 2011*; *Biter et al., 2012*).

The ClpB coiled-coil MDs were separately docked in the rod-shaped densities surrounding the AAA-1 ring (*Figure 1D*), maintaining the connection to the AAA-1 small subdomain. The pseudo-atomic model of one ClpB subunit obtained from this fitting differs from the crystal structure by rotations about the inter-domain hinge points (*Figure 1—figure supplement 4B*).

In this position, the MD contacts the neighbouring AAA-1 via its motif 1, while motif 2 makes intrasubunit AAA-1 interactions. This is in good agreement with recent biochemical data showing protection of these two motifs upon ClpB oligomerisation and formation of an intermolecular disulphide bond between E344C of AAA-1 and L424C of a neighbouring MD motif 1 (*Oguchi et al., 2012*; *Figure 1E*).

Moreover, intramolecular disulphide cross-links engineered between AAA-1 and motif 2 residues in *Tth*ClpB, *Eco*ClpB and yeast Hsp104 are also compatible with the fitted structure (*Figure 1E*; K476C/E358C, *Oguchi et al., 2012*; G175C/R484C, H362C/Q473C *E. coli* numbering, *Lee et al., 2003*; G175C/S499C, *Haslberger et al., 2007*; Hsp104 K358C/D484C, *Lipinska et al., 2013*). However, this arrangement is not compatible with some of the engineered disulphide bonds observed in yeast Hsp104 (*Desantis et al., 2014*) (D427C/E475C, D427C/E471C and E320C/N467C).

To investigate possible species-specific structural differences we reconstructed the functional yeast homologue HAP (Hsp104 with the ClpA tripeptide for ClpP binding) in complex with ClpP. Using ~10,000 particles and analyzing the data as described above, an independent map of HAP was obtained at ~21 Å resolution (*Figure 2A*, *Figure 1—figure supplement 3*). The three-layered structure is comparable to BAP and there is density accounting for all domains including the MD, which surrounds the AAA-1 ring. The atomic coordinates of the ND and AAA+ rings obtained from the BAP analysis were fitted as rigid bodies. In order to fit the density, the MD must adopt a more horizontal orientation (*Figure 2A*). In summary, HAP shows overall the same structural organisation as its bacterial homologue ClpB.

**Table 1.** X-ray data collection and refinement statistics

| Protein | ClpB E279A/E432A/E678A (SeMet) + ATP |
|---|---|
| Wavelength (Å) | 0.9794 |
| Space group | P6₅ |
| Unit cell (Å,°) | 127.34, 127.34, 119.86, 90, 90, 120 |
| Molecules | 1 |
| Resolution (Å) | 81.15–3.50 (3.69–3.50) |
| Reflections measured | 202965 |
| Unique reflections | 14024 |
| Rmerge | 8.8 (53.2) |
| Rpim | 3.0 (14.9) |
| I/σI | 16.8 (5.4) |
| Completeness (%) | 99.9 (100) |
| Redundancy | 14.5 (14.8) |
| Rwork/Rfree (%) | 22.5/24.5 |
| Protein residues | 657 (5257 atoms) |
| ADP | 2 |
| Rmsd bond lengths (Å) | 0.004 |
| Rmsd bond angles (°) | 0.94 |
| Average B-Factor Protein | 99.6 |
| Average B-Factor ADP | 87.7 |
| Ramachandran plot statistics | |
| Favored (%) | 95.5 |
| Allowed (%) | 4.1 |
| Generous (%) | 0.4 |
| Disallowed (%) | 0 |
| PDB entry code | 4CIU |

## Cryo EM reconstructions of ClpB with and without ClpP support the negative stain maps

Since negative stain EM of ClpB (BAP) in complex with ClpP gave a clear result different from all previous cryo EM maps of ClpB and Hsp104, we collected cryo EM data on BAP-ClpP as well as on ClpB alone. The same strategy of using only clearly identifiable side views was applied. Complexes were imaged in the presence of ATPγS and independent maps were obtained by de novo angular reconstitution in each case (*Figure 2B,C*).

For cryo EM of the BAP-ClpP complex, we used a Trap (E279A/E678A) variant, which can bind but not hydrolyze ATP due to mutations in both Walker B motifs. We anticipated that the Trap construct would be more stable than the wild type, but data collection was challenging because side views were not abundant. Eventually, ~4500 particles were collected and the same processing strategy was used as for the negative stain data. This variant is more likely to trap non native substrates in its central channel, and the extra density seen in this complex is likely to arise from denatured protein, possibly ClpP, present in the sample (*Figure 2B*).

In the case of isolated ClpB, side views were sorted on the basis of multivariate statistical analysis (MSA). Briefly, particles representing all views of ClpB were picked, centered and classified by MSA. Only particles belonging to classes representing side-views (80°< β <100°) were extracted and used for further processing.

Both maps show overall dimensions and density distributions comparable to the negative stain structure of BAP in complex with ClpP, confirming that the ClpB hexamer structure is not significantly altered either by negative staining or by binding to ClpP. The atomic model derived by fitting the negative stain BAP-ClpP map could be docked as a rigid body into both cryo EM reconstructions, but the MD assumes a more horizontal orientation, similar to that in HAP (*Figure 2*).

## ClpB activity mutants show altered MD orientations

E432A and Y503D are ClpB point mutations at opposite ends of the MD coiled-coil, which result in repressed (E432A) and hyperactive (Y503D) states (*Oguchi et al., 2012*; *Seyffer et al., 2012*). Repressed ClpB-E432A is deficient in DnaK interaction and cannot be activated by its Hsp70 partner. Hyperactive ClpB-Y503D shows high ATPase and substrate unfolding activity even in the absence of Hsp70 (*Oguchi et al., 2012*; *Seyffer et al., 2012*). We collected negative stain EM datasets of the BAP-ClpP complexes of these variants and obtained 3D reconstructions of BAP-E432A and BAP-Y503D at 18 Å and 20 Å, respectively (*Figure 3*, *Figure 1—figure supplement 3*). Starting models were independently generated by angular reconstitution and refined with sixfold symmetry (*Figure 3—figure supplement 1*). Both mutants assemble into three layers similar to the wild type and show high variability in the ND ring. Atomic coordinates of the AAA+ rings can be fitted as described for the wild type, using the ClpC hexamer as starting point. Some rearrangement was necessary to fit the ND into density (*Figure 3*). The most notable difference between the two maps is a ~30° difference in orientation of the motif 1 blade of the MD and the loss of the motif 2 density in the hyperactive state (*Figure 4A,B*). Another difference observed upon alignment of the AAA-1 ring in the

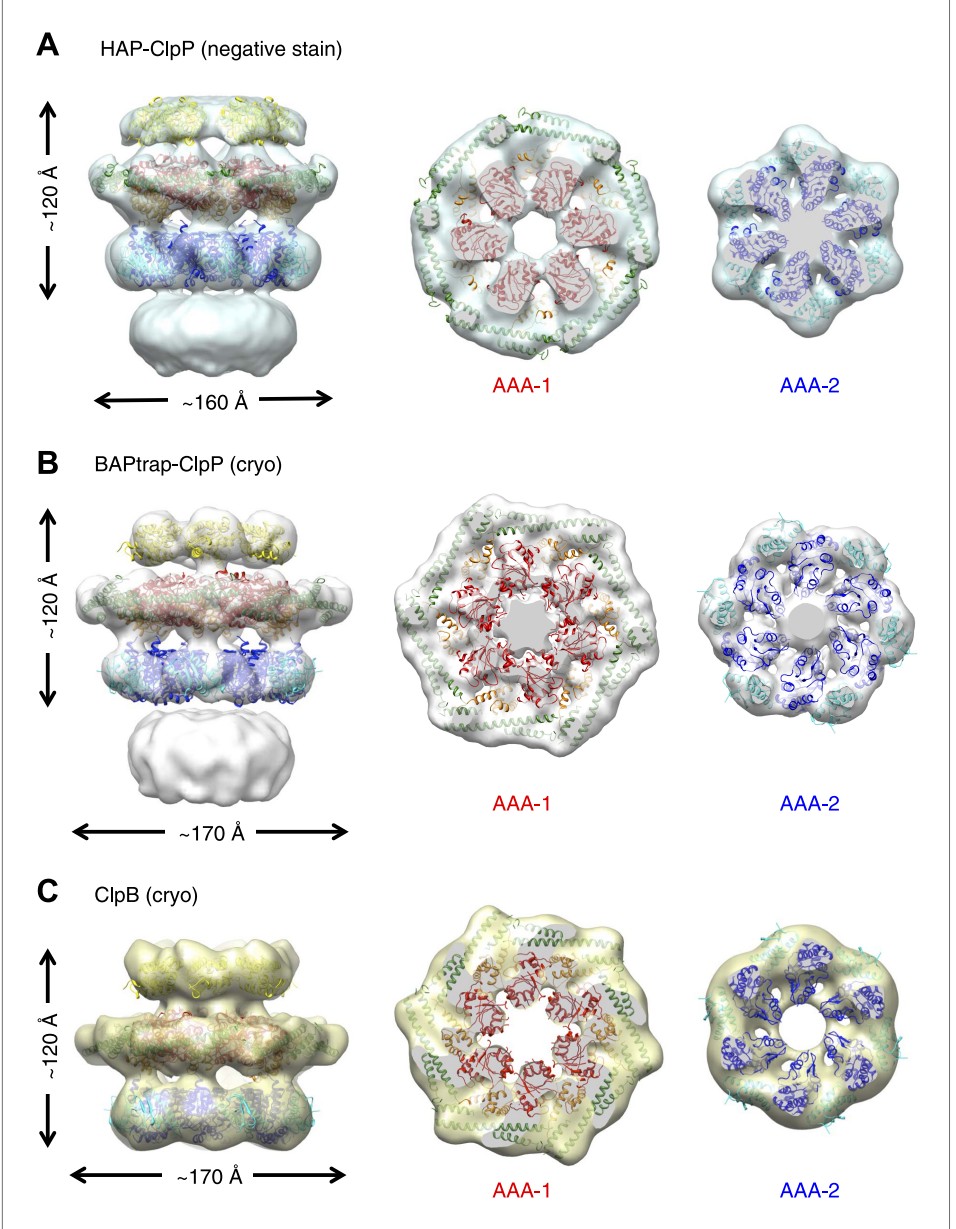

**Figure 2**. Independently determined maps and fitted hexameric models of HAP-ClpP, BAPtrap-ClpP and ClpB. (**A**) Negative stain EM map of HAP-ClpP. From left to right: surface side-view, AAA-1 layer and AAA-2 layer. The central channel enclosed by the AAA-2 ring is filled with density. (**B**) Cryo EM map of BAP-ClpP formed with the BAP variant that traps the substrate inside. (**C**) Cryo EM map of wild type ClpB alone.

repressed, wild-type and hyperactive maps is a ~15° rotation of the wild-type AAA-2 ring relative to the mutants.

In the BAP-E432A repressed mutant the MD is clearly visible around the AAA-1 ring. It has a horizontal orientation so that motif 2 forms a contact with motif 1 of the neighbouring MD (*Figure 3B*, *Figure 4B*). This packing of the MD is supported by biochemical data showing overprotection of motif 2 in this mutant (*Oguchi et al., 2012*). Moreover, a very similar intersubunit motif 1-motif 2 contact is seen in all the ClpB crystal structures, in which ClpB monomers are arranged in a spiral assembly (*Tth*ClpB, 1QVR, *Lee et al., 2003*; *Eco*ClpB current study) (*Figure 4—figure supplement 1*).

In the Y503D hyperactive form there is density for motif 1 in a more tilted orientation, but there is no density visible for motif 2, suggesting that this region becomes either disordered or highly mobile

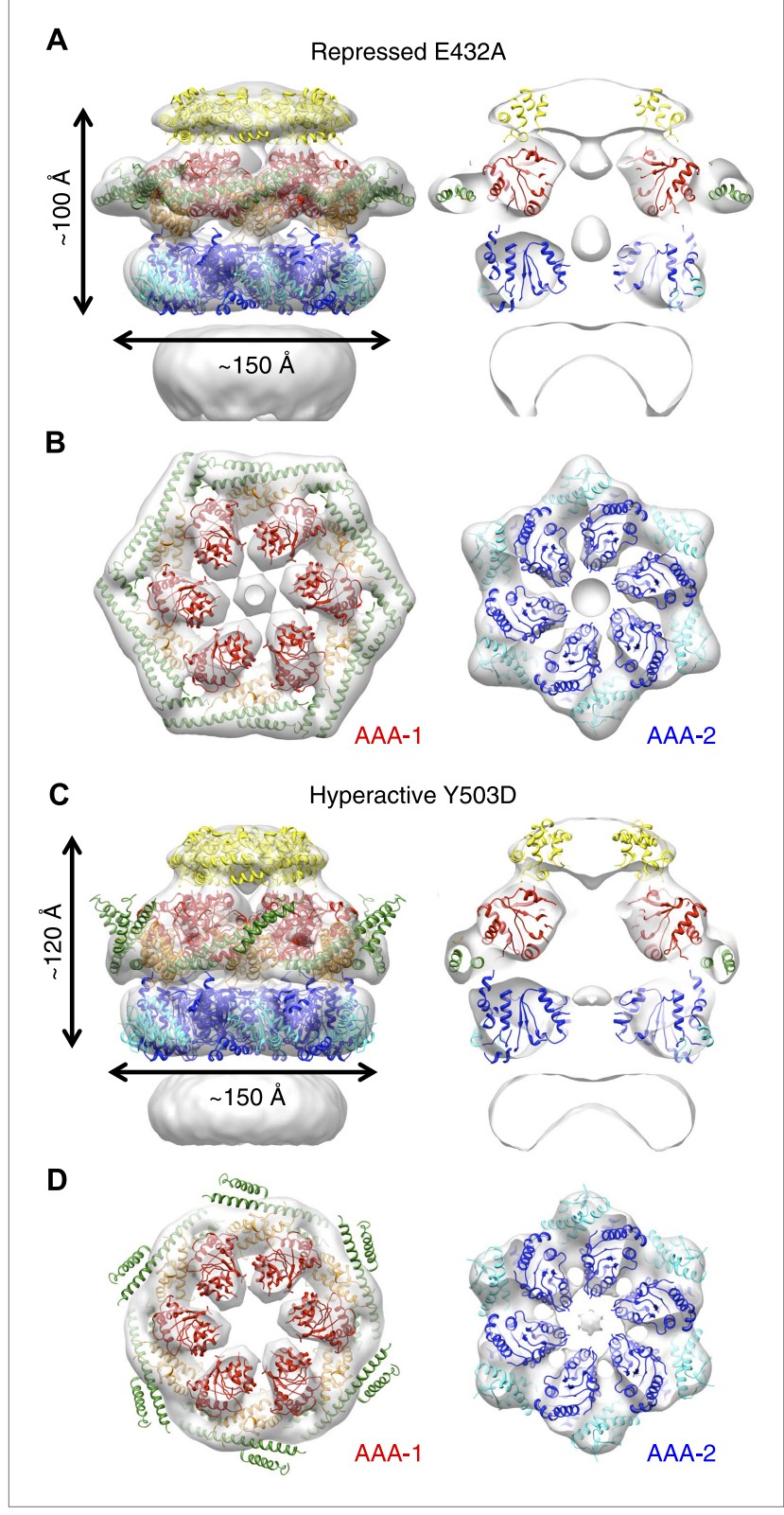

**Figure 3**. Structures of repressed and hyperactive BAP-ClpP mutants. (**A**) Side view (left) and 35 Å section (right) of the BAP-E432A map with the ClpB hexamer fit. (**B**) Fitting of the BAP-E432A AAA+ rings. (**C**) Side view (left) and 35 Å section of the fitted BAP-Y503D structure. MD is only partially in density and is more tilted than in the wild type and the BAP-E432A repressed mutant. The ND is smeared out as in all BAP forms due to disorder, but has a

*Figure 3. Continued on next page*

*Figure 3. Continued*

more vertical orientation in the hyperactive state. (**D**) Fitting of the BAP-Y503D AAA+ rings. The motif 2 region protrudes from the density.

The following figure supplements are available for figure 3:

**Figure supplement 1**. Test of EM map refinements after interchange of starting models.

(*Figure 3C,D*). This finding is in accordance with H/D exchange data showing deprotection of this region in the BAP-Y503D mutant (*Oguchi et al., 2012*). We conclude that, in the Y503D hyperactive form, the motif 1 arm of the MD is tilted downwards so that it can no longer contact motif 2 of the adjacent MD, and binds to a lower region of the neighbouring AAA-1 domain. Motif 2 is mobile and solvent exposed (*Figure 3C,D*, *Figure 4A,B*).

Alignment of AAA-1 domains of wild type, repressed and hyperactive forms of ClpB/BAP confirms that the MD rotates ~30° from a tilted orientation in the hyperactive state, to a horizontal one in the repressed state, with the wild type occupying an intermediate orientation (*Figure 4B*). In this movement motif 1 switches from being protected against AAA-1 to engaging motif 2 in trans. These different orientations of the MD have opposite consequences for DnaK binding. The horizontal MD position, with motif 1 contacting motif 2 of the neighbouring subunit, is incompatible with DnaK binding at the tip of motif 2 (*Seyffer et al., 2012*; *Rosenzweig et al., 2013*). In contrast, tilting of the MD exposes motif 2 for DnaK interaction.

To confirm the close proximity of motif 1 and motif 2 of neighbouring MDs we performed fluorescence resonance energy transfer (FRET) experiments using Q427W (motif 1) as FRET donor and IAEDANS labeled Q502C (motif 2) as FRET acceptor. The FRET pair (Förster radius of 22 Å) has a distance of 19.1 Å in the wild type model and was introduced into the tryptophan-free ClpB-W462Y/W543L variant. In addition we coupled the FRET pair to E432A and Y503D mutations, to monitor the consequences of repressed and hyperactive states on FRET efficiency. The distance between the FRET partners is either reduced (11.3 Å) or increased (35.9 Å) in models of the repressed and hyperactive variants, respectively (*Figure 4B*). All ClpB variants were IAEDANS labeled with similar efficiencies (70–80%; *Figure 4—figure supplement 2*). High IAEDANS fluorescence and thus FRET efficiency was observed for wild type ClpB and E432A upon ClpB oligomerization. Increase in acceptor fluorescence in general correlated with reduced tryptophan emission except for ClpB wild type upon oligomerization. In contrast, IAEDANS fluorescence remained low under all conditions tested when the FRET pair was linked to Y503D (*Figure 4C*).

Furthermore, to test for direct interaction between motif 1 and motif 2, we introduced cysteine residues into motif 1 (K431C) and motif 2 (S499C) and analyzed whether intermolecular crosslinks form under oxidizing conditions. Low Cu-Phenanthrolin concentrations (25 µM) yielded a ladder of crosslink products from ClpB dimers to hexamers. Although non-specific dimer formation was observed in single cysteine variants, higher oligomers were only found in double cysteine variants (*Figure 4D*). Introducing the activating (Y503D) mutation to ClpB-K431C/S499C decreased but did not abolish crosslink formation, suggesting rapid fluctuation of MDs between different conformations even in the hyperactive state (*Figure 4D*). These findings support the interaction of MD motifs 1 and 2 from adjacent subunits as observed in the EM reconstructions and strengthening or loosening of the contact in repressed and hyperactive states, respectively.

## Motif 1 is essential for keeping ClpB in the repressed state

Mutating MD motif 1 can impair Hsp70 interaction and consequently protein disaggregation (*Oguchi et al., 2012*; *Seyffer et al., 2012*). On the other hand a ClpB motif 1 deletion variant retains substantial disaggregation activity (*Seyffer et al., 2012*; *del Castillo et al., 2011*), providing no clarification of its role in protein disaggregation. Our EM structural data indicate that motif 1 acts as a crucial component of ClpB regulation by stabilizing MDs in a horizontal position through interaction with an adjacent motif 2. Indeed, ClpB-ΔM1 (ΔE410-Y455) shows hyperstimulation of ATPase activity in the presence of the substrate casein compared to wild-type ClpB, resembling the deregulated ATPase activity of hyperactive ClpB variants lacking motif 2 or the entire MD (*Oguchi et al., 2012*; *Figure 5A*). The basal ATPase activity of ClpB-ΔM1 was reduced in comparison to wild type due to partial

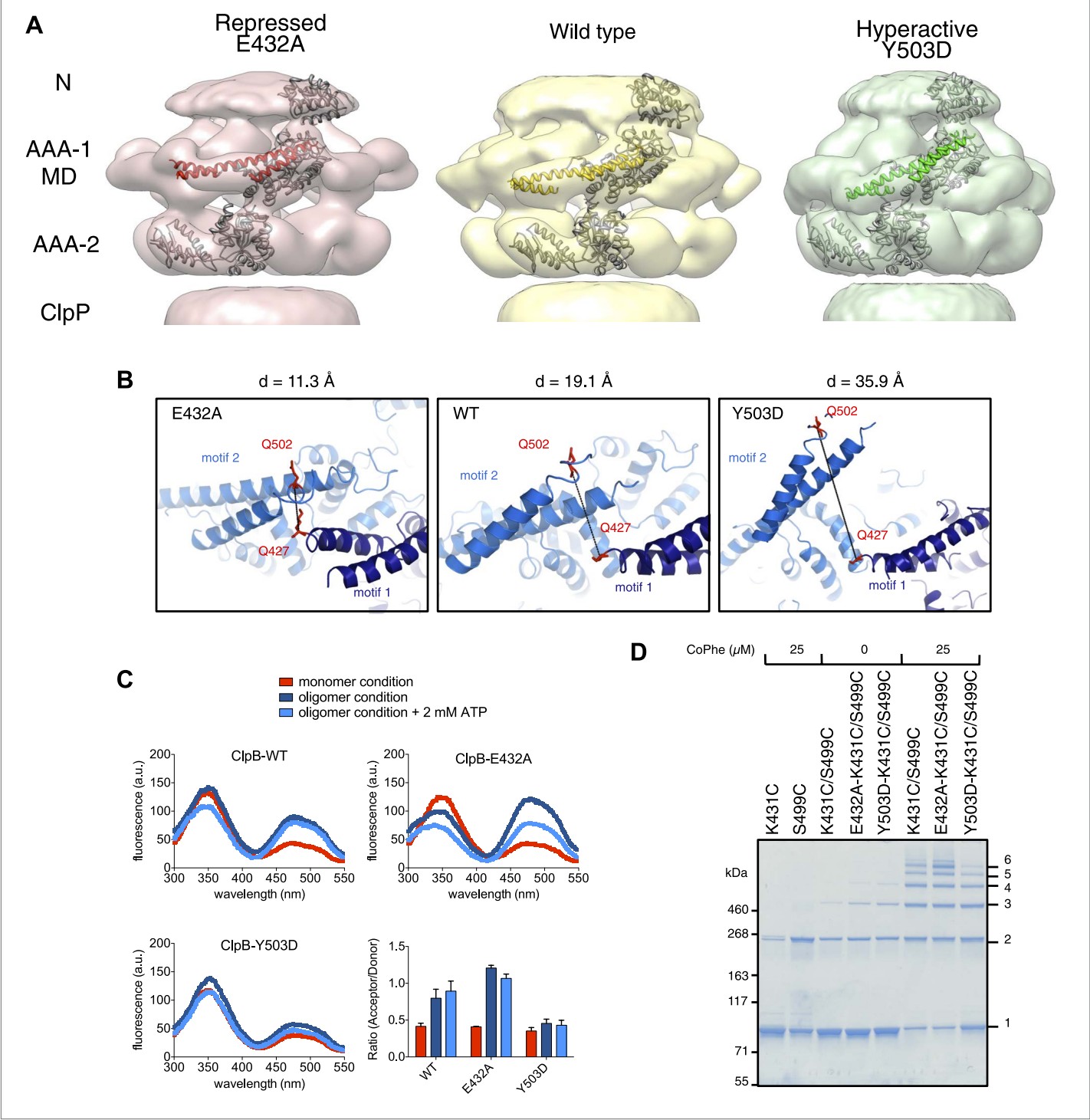

**Figure 4**. Analysis of MD movements. (**A**) Repressed (red), wild-type (yellow) and hyperactive (green) forms of BAP-ClpP with a fitted subunit. (**B**) Distances (**D**) are shown between Cα atoms of Q427 (motif 1) and Q502 (motif 2) of neighbouring subunits based on the symmetrised EM reconstructions of wild-type ClpB and MD mutants. (**C**) Fluorescence energy transfer between motif 1 and motif 2 of adjacent ClpB subunits. Emission spectra of ClpB-Q427W-Q502C-IAEDANS in the monomeric (high salt), oligomeric (low salt) and ATP state (2 mM ATP) are shown. Effects of MD mutations on FRET efficiency were determined. The ratio of acceptor to donor fluorescence (derived from the areas under the curves) was calculated as readout for FRET efficiency. (**D**) Intermolecular disulfide crosslinking between MD motif 1 and motif 2 of adjacent subunits. Reduced and oxidized ClpB-K431C-S499C and repressed (E432A) and hyperactive (Y503D) variants were analyzed by non-reducing SDS-PAGE. Positions of monomers (1) and

*Figure 4. Continued on next page*

*Figure 4. Continued*

crosslinked dimers (2), trimers (3), tetramers (4), pentamers (5) and hexamers (6) are indicated. A protein standard is given. CoPhe at 25 µM was used as oxidizing agent.

The following figure supplements are available for figure 4:

**Figure supplement 1**. Contacts between adjacent MDs in the EM model compared to those found in the ClpB crystal structures.

**Figure supplement 2**. Tryptophan emission spectra of unlabeled ClpB-Q427W-Q502C and of corresponding MD mutants in the monomeric (HS, high salt), oligomeric (LS, low salt) and ATP-loaded state (LS + 2 mM ATP) are given.

oligomerization defects, in agreement with earlier reports (*Oguchi et al., 2012*; *del Castillo et al., 2011*). ClpB-ΔM1 oligomerization defects were more pronounced than in ClpB-ΔM1/M2, suggesting that motif 2 in absence of motif 1 might impede hexamer formation.

Next we tested whether hyperstimulation of ATPase activity is linked to high unfolding power. We employed BAP variants of the respective deletion constructs and tested for unfolding and degradation of casein-YFP in presence of ClpP. BAP-ΔM1 unfolds the YFP moiety of casein-YFP, an activity only observed for hyperactive BAP variants but not wild type BAP (*Oguchi et al., 2012*; *Figure 5B*). Loss of ClpB regulation in ClpB-ΔM1 was also linked to severe toxicity upon expression in *E. coli* Δ*clpB* mutant cells (*Figure 5C*). However, higher expression levels of ClpB-ΔM1 than of ClpB-Y503D are required to observe the same degree of toxicity probably because deletion of motif 1 results in oligomerisation defects (*Oguchi et al., 2012*; *Figure 5C*). In conclusion ClpB-ΔM1 exhibits the three major characteristics of hyperactive ClpB variants (high ATPase and unfolding activities linked to cellular toxicity), demonstrating the crucial regulatory role for motif 1 in interacting with an adjacent motif 2 to ensure tight activity control of ClpB.

## Asymmetric reconstructions show variable orientations of the MD around the ring

As mentioned above, we imposed 6-fold symmetry as a first approximation, to simplify the alignment and reconstruction problem. Nevertheless, crystal structures of the AAA+ protein ClpX show that the homo-hexameric assembly can be markedly asymmetric (*Glynn et al., 2009*; *Kon et al., 2012*; *Stinson et al., 2013*). We therefore reanalysed our negative stain EM data without imposing symmetry, in order to study the conformational variability within the hexamer.

Using ~15,000, ~10,000 and ~9000 particles for wild-type BAP, BAP-E432A and BAP-Y503D we obtained asymmetric maps at 21 Å, 21 Å and 25 Å resolution, respectively (*Figure 6A*, *Figure 1—figure supplement 3*, *Figure 6—figure supplement 1*). Although the numbers of particles were similar to those used for sixfold analysis, the resolution is only slightly worse and the maps are comparable in quality to the symmetrised ones, consistent with the structure being asymmetric. The structure of the hyperactive mutant is less defined and has lower resolution than the other two, particularly for the MDs, probably owing to their higher mobility. Therefore, only wild-type BAP and BAP-E432A were used for further analysis.

The asymmetric structures show similar density distributions in the AAA+ rings, even though not all six subunits in a ring can be aligned simultaneously owing to 5°–15° variations in rotational orientation. This rotational variability may explain the apparent ~15° rotation of the AAA-2 wild-type ring relative to the mutants observed in the symmetrised reconstructions. As expected from the eigenimage analysis, the ND ring is not very well defined and the density does not account for all six NDs. Therefore, we did not attempt any atomic structure fitting into this region. Docking of the atomic structures of AAA-1, AAA-2 and MD was performed as follows.

Structures of AAA+ proteins crystallised in their hexameric assemblies show that the large and small subdomains of the AAA fold assume a range of conformations that can be clustered into open or closed forms (*Cho and Vale, 2012*; *Stinson et al., 2013*; *Figure 6—figure supplement 2*). The closed conformation is ATP-binding competent and there are intermediate forms that might also represent weak binding states. Using a gallery of available AAA+ crystal structures we modelled open, closed and intermediate conformations of ClpB AAA-1 and AAA-2. We also created AAA+ dimers of adjacent ClpB subunits based on ClpX pseudo-hexameric crystal structures (*Glynn et al., 2009*;

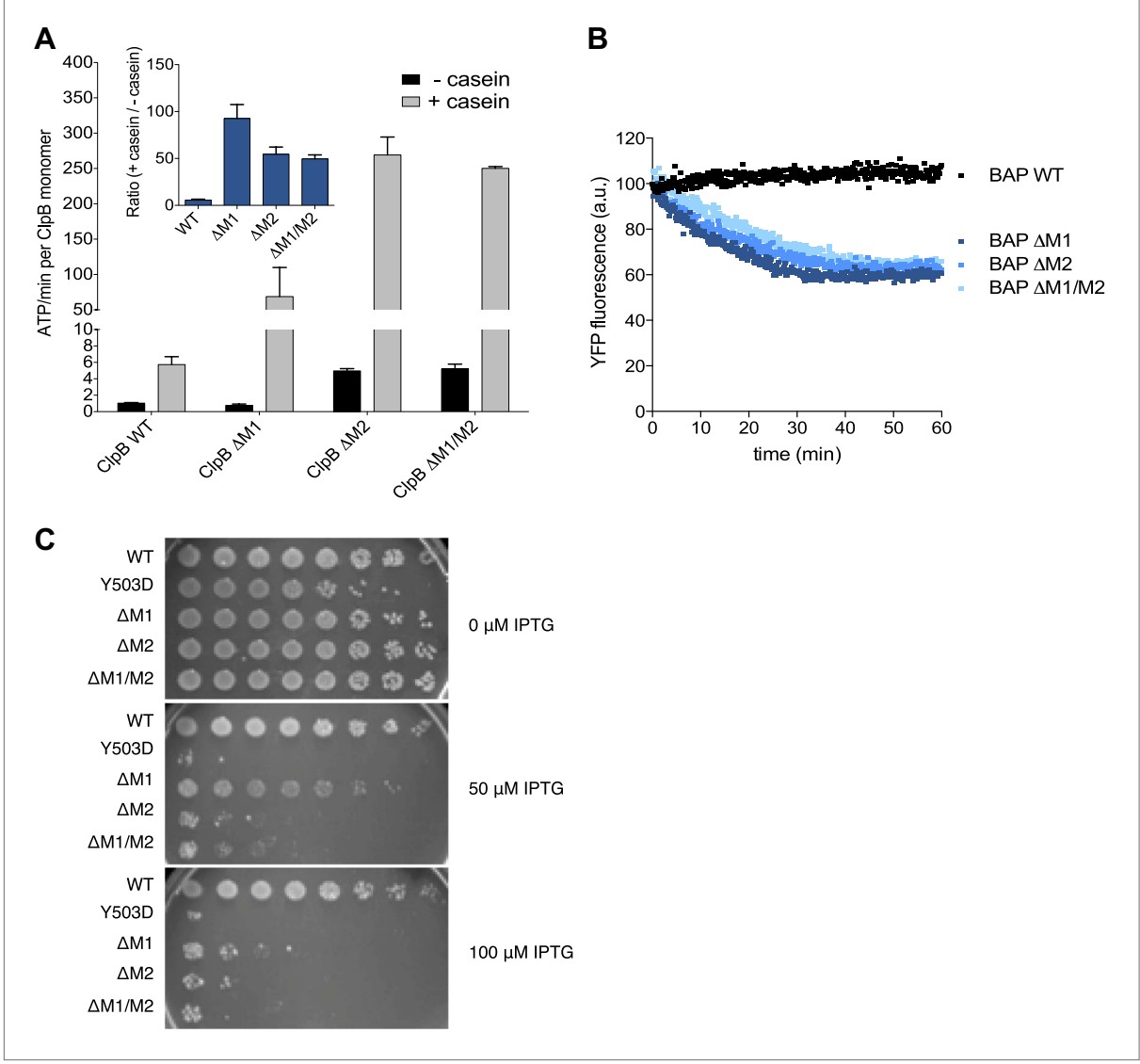

**Figure 5**. Deletion of MD motif 1 causes ClpB activation. (**A**) Basal and substrate-stimulated ATPase activities of ClpB wild type and indicated MD deletions were determined in the absence and presence of casein. Relative ATPase activations by casein were calculated (inset). (**B**) Unfolding of Casein-YFP by BAP wild type and indicated MD deletions in the presence of ClpP was monitored by YFP fluorescence. Initial YFP fluorescence was set to 100%. ΔM1: ΔE410-Y455, ΔM2: ΔS456-E520, ΔM1/M2: ΔE410-E520. (**C**) *E. coli* Δ*clpB* cells expressing the indicated plasmid-encoded *clpB* alleles under control of an IPTG-regulatable promoter were grown overnight at 30°C. Various dilutions (10⁻¹–10⁻⁷) were spotted on LB plates containing the indicated IPTG concentration and incubated at 37°C for 24 hr.

*Stinson et al., 2013*; *Figure 6—figure supplement 2*) that were fitted as rigid bodies into the asymmetric reconstructions. Crystallographic dimers are likely to provide more realistic models of subunit interfaces than can be deduced by fitting individual subunits into low-resolution maps.

The AAA-2 ring density could be almost entirely interpreted using this approach and the fit suggests that 3 to 5 subunits are sufficiently closed to bind ATP (*Figure 6—figure supplement 2*). We analyzed nucleotide (ADP) binding to wild type ClpB by isothermal calorimetry (ITC) revealing a binding stoichiometry of 7.5 ± 0.1 ADP per hexamer (*Figure 6—figure supplement 3*). ITC experiments using ClpB-K212A, which is deficient in nucleotide binding in AAA-1, allowed us to determine a binding stoichiometry of 3.7 ± 0.3 ADP in AAA-2 of the mutant hexamer (*Figure 6—figure supplement 3*). The same ADP binding stoichiometry was also found for the repressed and hyperactive variants (*Figure 6—figure supplement 3*). The deduced stoichiometries are in good agreement with the distinct AAA-2 conformations observed in the asymmetric EM reconstructions. Similar calculations have been reported

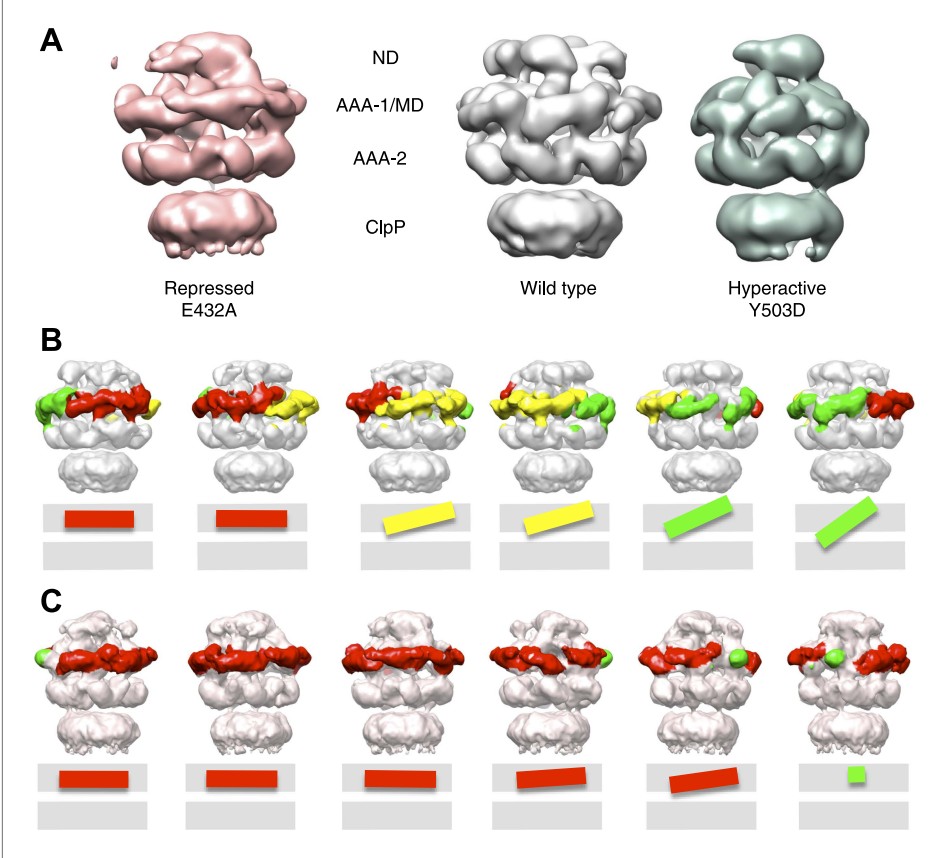

**Figure 6**. Variations in MD orientations around the ring. (**A**) Asymmetric reconstructions of repressed BAP-E432A, wild type BAP-ClpP and hyperactive BAP-Y503. (**B**) Side views of the BAP-ClpP wild type structure rotated successively 60°. Red MDs are oriented horizontally and make motif 1 to motif 2 contact as in the repressed state. Yellow MDs have slightly tilted orientations similar to the wild type. Green MDs are tilted as in the hyperactive state. One of the MDs contacts AAA-2 of the same subunit (last right panel). (**C**) Equivalent views of the BAP E432A-ClpP repressed mutant structure. Color code for the MD orientations is the same as above.

The following figure supplements are available for figure 6:

**Figure supplement 1**. Plots showing the angular distribution of single particles (red dots) around the Euler sphere for each asymmetric reconstruction.

**Figure supplement 2**. Open and closed conformations of AAA+ domains can be fitted into the asymmetric ClpB map.

**Figure supplement 3**. Asymmetric nucleotide binding to ClpB.

for the AAA+ proteins MCM, ClpX and HslU (*Moreau et al., 2007*; *Yakamavich et al., 2008*). The AAA-1 ring is more asymmetric than AAA-2 and it is not easily interpretable by fitting crystallographic dimer models. There is sufficient density to guide rigid body fitting of all AAA-1 domains, but this fitting does not allow deductions of nucleotide occupancy (*Figure 6—figure supplement 2*).

Asymmetric reconstructions of both wild-type BAP and repressed mutant display clear densities accounting for the MDs that lie outside the AAA-1 ring and assume different orientations similar to those observed in the symmetrised maps of wild type and mutants. In the wild-type asymmetric reconstruction, the MD orientation ranges from horizontal as in the repressed state to highly tilted, similar to the hyperactive state, passing through the intermediate wild type-like state (*Figure 6B*). In one subunit, motif 1 contacts AAA-2 of the same subunit, suggesting a route of allosteric communication between AAA-1 and AAA-2. Additionally, this contact is compatible with recently reported engineered disulphide bonds between motif 1 residues and AAA-2 in Hsp104 and ClpB (*Desantis et al., 2014*).

The tilted MDs are clustered together, consistent with the release of motif 1-motif 2 contacts freeing adjacent subunits.

In the asymmetric reconstruction of the repressed state, five MDs are found in the horizontal orientation, followed by a sixth for which there is no clear density (*Figure 6C*). In the wild type two MDs can make the motif 1 to motif 2 contact, while in the repressed state up to five MDs are compatible with this contact (*Figure 6B,C*). Conversely, in the wild type at least two MDs exist in a clearly activated state but in the repressed state none of the MDs are visible in the tilted, hyperactive conformation. As a consequence, the MD conformations in BAP-E432A do not support Hsp70 binding and ATPase activation, in agreement with previous findings (*Oguchi et al., 2012*; *Seyffer et al., 2012*). However, ClpB wild type hexamers harbour two MDs that favour Hsp70 recruitment, potentially priming the ClpB ring for further activation.

## Discussion

The BAP construct of ClpB complexed with ClpP, combined with a conservative approach of using only clearly identifiable side views and basing the analysis mainly on negative stain images (with independent confirmation from cryo EM) enabled us to unambiguously locate all subunit domains in the oligomer, including the coiled-coil MD propeller. Although the quoted resolutions for some previous structures of ClpB and Hsp104 were better, the globular shape, flexibility and asymmetry of these hexamers reduce the reliability of orientation assignment. Moreover, previous published structures of ClpB/Hsp104 were obtained by cryo EM only, which provides a lower signal-to-noise ratio (SNR) than negative stain data.

Rigid body fitting of *E. coli* ClpB atomic coordinates (PDB code 4CIU) into our new maps reveals that the MD is not projecting outwards from the hexamer (*Lee et al., 2003*, *2007*, *2010*) nor intercalated between subunits (*Wendler et al., 2007*, *2009*), but is instead lying on the surface of the ClpB hexameric ring with a variable degree of tilt (*Figure 4A*, *Figure 6B,C*). A similar position of the MD is seen for Hsp104, underlining the conserved activity and mechanism of these disaggregases. This new MD arrangement is in excellent agreement with recent, extensive biochemical analysis of the MD (*Oguchi et al., 2012*). It was proposed that the MD works as a molecular toggle switching from a repressed state in which both motif 1 and 2 ends of the propeller are protected, to an active state where motif 2 is deprotected, exposing the binding site for DnaK/Hsp70 (*Oguchi et al., 2012*; *Seyffer et al., 2012*; *Rosenzweig et al., 2013*). Our EM reconstructions of repressed, hyperactive and wild type ClpB reveal a lever-like movement of the MD that switches from the repressed state with head-to-tail motif 1-motif 2 binding between adjacent subunits, to a mobile, activated state with motif 2 free and available for binding to DnaK (*Figure 7A*). The wild type conformation is intermediate between these states and thus poised for switching.

Our data thus explain the critical role of motif 1, which regulates the accessibility of motif 2 and consequently ClpB activity. Confirming its important regulatory role, deletion of motif 1 results in hyperactive ATPase and unfolding activity of ClpB-ΔM1 as well as cell toxicity (*Figure 5*), representing key characteristics of the hyperactive state (*Oguchi et al., 2012*). Moreover, our maps suggest that motif 1, through its contacts with either the adjacent motif 2 or AAA-1, plays a key role in direct communication between neighbouring subunits, which must therefore act in a coordinated manner.

It has become clear that AAA+ proteins are highly dynamic molecular motors unlikely to exist in a homogeneous structural state. Therefore we generated asymmetric reconstructions of ClpB. Although the resolution of the asymmetric structures is not sufficient to support a detailed mechanistic model, these reconstructions provide the first visualization of the MD conformational flexibility that was inferred from biochemical analysis (*Lee et al., 2003*; *Haslberger et al., 2007*; *Oguchi et al., 2012*). The asymmetric structures show that the MD orientation varies around the ring occupying the repressed, wild type-like and hyperactive positions described by the symmetrised averages. The variable tilts of MDs observed around the ring suggest that 2 to 4 adjacent subunits are available for DnaK binding in the wild-type vs only 1 in the repressed mutant (*Figure 6B,C*). This is consistent with the estimated stoichiometry of 2–5 molecules of DnaK per ClpB hexamer required for activation (*Seyffer et al., 2012*; *Desantis et al., 2014*). It also suggests that at least four subunits must have detached MDs to allow activity, perhaps through movements of the AAA+ domains, in agreement with the number of ClpX subunits that hydrolyze ATP in a coordinated manner to unfold GFP in single molecule experiments (*Sen et al., 2013*). Moreover, we calculated an ADP binding stoichiometry of 4 for both wild type and mutants (*Figure 6—figure supplement 3*), which indicates that although ATP hydrolysis is strongly affected, detachment of the MD does not change the nucleotide binding.

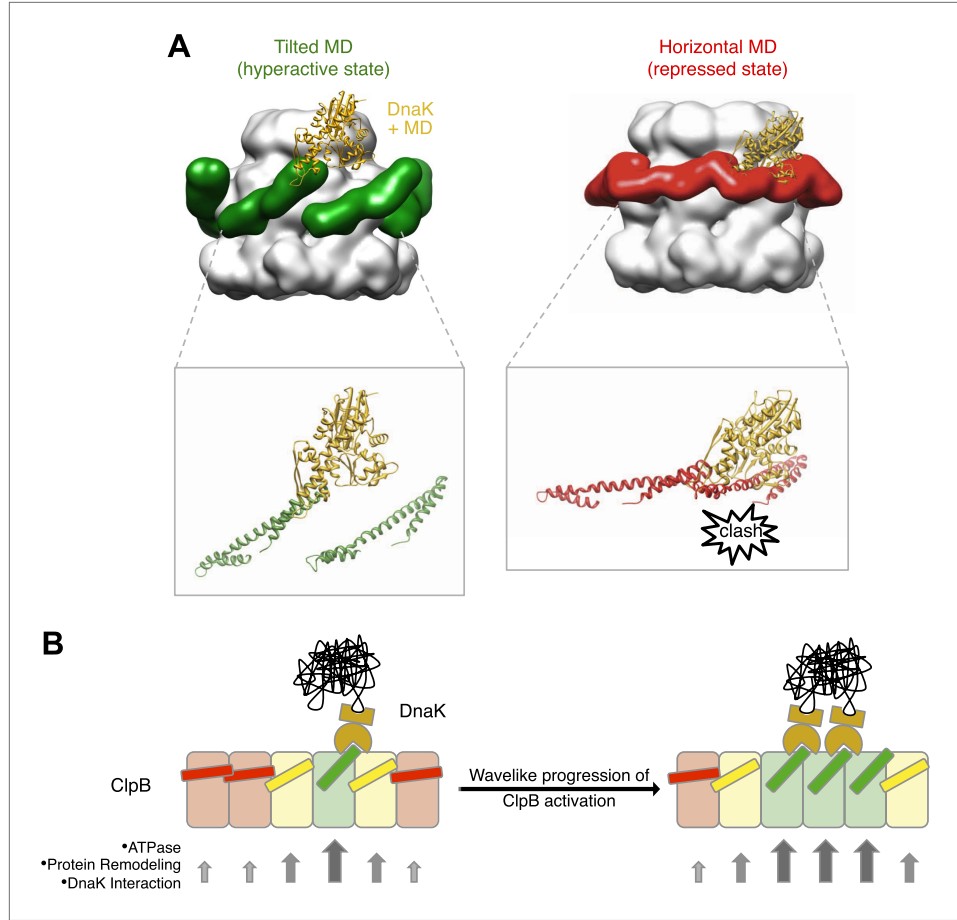

**Figure 7**. Interaction between the ClpB hexamer and DnaK. (**A**) Binding of DnaK is only possible when the MD is tilted (green). DnaK cannot bind when two adjacent MDs occupy a more horizontal position (red) as it would clash with motif 1 of the neighbouring ClpB subunit. (**B**) Cartoon of an opened-out ClpB ring with two DnaK molecules bound. The MDs involved in DnaK binding are shown in green, and those that cannot bind to DnaK are shown in red. Release of the ends of the MD favours activation of adjacent subunits, and the active cluster can move around the ring in a wave-like manner. The model of DnaK binding to the MD is based on the NMR study in *Seyffer et al. (2012)*.

The cryo EM reconstruction of wild-type ClpB suggests that the various MD conformations exist only transiently, poised between hyperactive and repressed states, with the balance shifted slightly towards repressive motif 1-motif 2 contacts (*Figure 2*). DnaK binding to an accessible motif 2 stabilizes the MD in a tilted conformation, thereby in turn breaking the repressive contacts with MDs in neighbouring subunits. Thus, an initial encounter of DnaK will facilitate DnaK binding in the neighbouring MDs. In this model, breakage or formation of motif 1-motif 2 contacts provides a mechanistic basis for signalling DnaK binding or dissociation in a wavelike manner around the ClpB ring. (*Figure 7B*). The model predicts spatial proximity of multiple DnaK molecules, which is the case for aggregated but not soluble DnaK substrates, directing ClpB activity specifically to protein aggregates. Moreover, the activation of ClpB by DnaK binding, combined with movements of the highly mobile N domain, might act to deliver the substrate to the channel entrance, where it would be engaged for threading, unfolding and consequent extraction from the aggregate.

## Material and methods

### Strains, plasmids and proteins

*E. coli* strains used were derivatives of MC4100. Mutant derivatives of ClpB/BAP were generated by PCR mutagenesis and standard cloning techniques in pDS56 and were verified by sequencing. Wild

type and mutant ClpB were purified using Ni-IDA (Macherey–Nagel) and size exclusion chromatography (Superdex S200, Amersham) following standard protocols. Purifications of DnaK, DnaJ, GrpE, ClpP, Luciferase and Casein-YFP were performed as described previously (*Oguchi et al., 2012*). Pyruvate kinase and α–casein were purchased from Sigma. Protein concentrations were determined with the Bio-Rad Bradford assay.

## Electron microscopy of negatively stained and vitrified specimens

BAP(HAP)-ClpP complexes were formed in 20 mM Tris–HCl, pH 7.5, 20 mM KCl, 15 mM MgCl$_2$, 1 mM DTT and 2 mM ATPγS. Proteins were applied to glow-discharged carbon coated grids (EM sciences), previously coated with 5 kDa poly-lysine (Sigma-Aldrich, UK) to positively charge the surface. Samples were stained with 2% uranyl acetate.

For cryo-EM imaging, BAPtrap-ClpP was applied to holey carbon grids coated with a thin carbon film and pretreated with poly-lysine while ClpB specimens were loaded onto lacey carbon grids. Cryo-EM specimens were vitrified in a Vitrobot (FEI, UK). Images were recorded on a 4k Gatan (UK) CCD camera at a magnification of 50,000X for negatively stained specimens (pixel size 2.2 E; underfocus range: 0.5–1.2 µm) and of 80,000 × for cryo specimens (pixel size 1.4 Å; underfocus range: 1.5 µM–4 µm). All data were collected on a Tecnai F20 FEG operated at 200 kV under low dose conditions.

## Single particle processing

The contrast transfer function (CTF) for each CCD frame was determined with CTFFIND3 (*Mindell and Grigorieff, 2003*) and corrected by phase flipping using Bshow1.6 (*Heymann and Belnap, 2007*). Side views of BAP-ClpP 2:2 complexes were manually picked using Boxer (*Ludtke et al., 1999*) and extracted into 256 × 256 boxes. The boxed particles were band-pass filtered between 300 and 10 Å for the negative stain dataset and between 300 and 5 Å for the cryo dataset. They were then normalized to the same mean and standard deviation. Particles were initially aligned to the total sum of 10–20 vertically oriented particles using SPIDER (*Frank et al., 1996*). Individual 1:1 BAP-ClpP complexes were extracted with circular masks and classified by MSA in IMAGIC-5 (*van Heel et al., 1996*) to remove images that did not represent BAP-ClpP, yielding 17,470 particles of wild type BAP-ClpP, 12,588 of BAP-E432A-ClpP, 9436 of BAP-Y503D-ClpP and 12,568 of HAP-ClpP. The BAPtrap-ClpP cryo dataset included 4592 particles. At this stage, particles were high-pass filtered to 160 Å and initial class averages of 20–30 images each were obtained by MSA. All alignments of 1:1 BAP-ClpP complexes were done limiting the in-plane rotation to ±20°. Upon further classification, 5–10 good classes were used to generate a starting model by angular reconstitution (*van Heel et al., 2000*). Alternative starting models were also created by applying sixfold symmetry to single classes and then using the resultant 3D map, generally composed of three discs corresponding to the three layers of the molecule, to generate an anchor set for Euler angle assignment. A low-resolution density map was independently created for each dataset by angular reconstitution with sixfold symmetry. Particle orientations were refined in multiple cycles of AP SHC alignment in SPIDER, MSA and angular reconstitution in IMAGIC and the resulting 3D reconstruction, filtered to 30 Å, was used as an initial model for projection matching in SPIDER. By applying a rectangular mask, only the AAA+ layers were refined. After 8–10 cycles of projection matching, using progressively smaller angular sampling steps (4°–1°) and filtering the 3D to the estimated resolution at each cycle, final structures were generated of the whole complex and their resolution was estimated by Fourier shell correlation with a 0.5 correlation cutoff. Based on cross correlation coefficient, around 70–80% of each dataset was included in the final reconstruction.

The differences between the mutant structures were tested by refining with interchanged starting models. In both cases, the original mutant structure was recovered despite the use of the other map as a starting model for projection matching (*Figure 3—figure supplement 1*).

For cryo EM of ClpB alone, the views were randomly oriented and the initial strategy was to extract clearly identifiable side views by MSA and classification. 7606 side views were used to generate a starting model by angular reconstitution, which was refined to 29 Å resolution (*Figure 1—figure supplement 3*) by projection matching.

For the reconstructions without imposed symmetry, particle orientations were determined using either the sixfold symmetrised map filtered to 50 Å or a sphere obtained from the average of all particles as a starting model. Subsequently, particle orientations were refined with ~10 cycles of projection matching without imposed symmetry. The particle orientations were well distributed around the

BAP-ClpP central axis (*Figure 6—figure supplement 1*). The resolution was estimated by Fourier shell correlation (FSC) at 0.5 (*Figure 1—figure supplement 3*).

## Atomic structure fitting

Docking was done using the crystal structures of *E. coli* ClpB-E432A (current study, PDB code: 4CIU) and *E. coli* ClpB ND (PDB code 1KHY). A homology model of Hsp104, obtained using Phyre2 (*Kelley and Sternberg, 2009*) was used for fitting into the HAP reconstruction.

A hexameric ClpB/Hsp104 AAA-1 ring was modelled on the ClpC hexamer crystal structure (PDB code 3PXG) and was automatically fitted as a rigid body into the symmetrised maps using the UCSF Chimera package (*Pettersen et al., 2004*). The N-terminal, the middle and the AAA-2 domains were first fitted manually and then local fitting was optimized in Chimera, followed by symmetrisation. In the asymmetric reconstructions, dimers of adjacent ClpB subunits were modelled on crystal structures of ClpX hexamers and fitted as rigid bodies in Chimera.

## Biochemical assays

Steady-state ATP hydrolysis rates were determined in buffer A (50 mM Tris pH 7.5, 5 mM $MgCl_2$, 20 mM KCl, 2 mM DTT) as described (*Haslberger et al., 2007*). ClpB disaggregation activities were determined by following the refolding of aggregated firefly Luciferase according to published protocols (*Haslberger et al., 2007*). Chaperones were used at the following concentrations: 1 µM ClpB (wild type or derivatives), 1 µM DnaK, 0.2 µM DnaJ, 0.1 µM GrpE. Oligomerisation of ClpB variants was tested as described previously (*Mogk et al., 2003*; *Haslberger et al., 2007*).

Site-specific labelling of ClpB using 1,5-IAEDANS, 5-([(2-iodoacetyl)amino]ethylamino)naphthalene-1-sulfonic acid (Invitrogen, Germany) was performed according to the manufacturers' instructions. The intrinsic tryptophan fluorescence of ClpB variant harbouring a single tryptophan (ClpB*-Q427CW, 1 µM each) was measured on a Perkin-Elmer (Germany) LS50B spectrofluorimeter at 25°C in low salt buffer A (50 mM Tris pH 7.5, 5 mM $MgCl_2$, 20 mM KCl, 2 mM DTT). Tryptophan and IAEDANS emission spectra of labeled ClpB variants were recorded in high and low salt buffer A (supplemented with 400 mM KCl or 20 mM KCl, respectively) in the absence or presence of 2 mM nucleotide between 300 and 550 nm at a fixed excitation wavelength of 290 nm. The Förster radius of the Trp-IAEDANS FRET pair was calculated as 22 Å (*Jeganathan et al., 2006*).

For formation of disulfide bridges ClpB cysteine variants were first dialyzed to remove DTT. Cysteine oxidation was achieved by adding 25 µM Cu-Phenanthroline to 4 µM ClpB and incubating the mixture for 1 min at room temperature. Oxidation and disulfide bond formation was stopped by addition of 50 mM iodoacetamide and SDS-sample buffer containing 5 mM EDTA. Crosslink products were analyzed by a non-reducing SDS gradient gel (3–8%).

Casein-YFP unfolding and degradation assays were carried out using 6 µM BAP (wild type or variants), 9 µM ClpP and 0.5 µM Casein-YFP. Degradation of Casein-YFP was determined by monitoring YFP fluorescence at 535 nm (excitation wavelength 515 nm) at a Perkin–Elmer LS50B spectrofluorimeter.

## Spot tests

*E. coli* cells harbouring plasmid-encoded clpB alleles were grown in the absence of IPTG overnight at 30°C. Serial dilutions were prepared, spotted on LB-plates containing different IPTG concentrations and incubated for 24 hr at 37°C.

## Isothermal titration calorimetry

ClpB wild type and variants were extensively dialyzed against low salt buffer A (50 mM Tris [pH 7.5], 25 mM KCl, 20 mM $MgCl_2$, 5% glycerol). Isothermal titration calorimetry (ITC) was performed using an ITC calorimeter (iTC200Microcalorimeter, MicroCal, Germany). Consecutive injections of nucleotide into a 300 µl cell containing ClpB were performed after sample equilibration at 30°C. Integration and fitting of ITC data were performed using ORIGIN software (GE, Germany). ClpB and ADP concentrations were determined by UV absorbance at 280 nm.

## Hydrogen-exchange experiments

HX-MS experiments were performed similar to those described earlier (*Oguchi et al., 2012*). Briefly, ClpB (100 pmol), BAP (100 pmol) or BAP-ClpP complex (100 pmol and 200 pmol respectively) were incubated for 3 min at 30°C in low salt buffer A (50 mM Tris, pH 7.5, 25 mM KCl, 20 mM $MgCl_2$, 2 mM DTT) in presence of ATP or ATPγS and diluted 20-fold into $D_2O$-based MDH buffer to initiate amide

proton-deuteron exchange. The exchange reaction was stopped after 1 min by the addition of 1 volume of ice-cold quench buffer (0.4 M K-phosphate buffer, pH 2.2). Quenched samples were immediately injected into the HPLC setup, with (peptide analysis) or without (full length protein analysis) online peptic digest, and analyzed on an electrospray ionization quadrupole time-of-flight mass spectrometer (QSTAR Pulsar, Applied Biosystems) as described in *Rist et al. (2003)*. Analysis of deuteron incorporation into peptide was performed by using AnalystQS software (Applied Biosystems/MDS SCIEX, Germany).

### Accession numbers

The EM maps have been deposited in the 3D-EM database (www.emdatabank.org) with accession codes EMD-2555 (BAP-E432A C6), EMD-2556 (BAP-E432A C1), EMD-2557 (BAP wild type C6), and EMD-2558 (BAP wild type C1), EMD-2559 (BAP-Y503D C6), EMD-2560 (BAP-Y503D C1), EMD-2561 (HAP), EMD-2562 (BAPtrap cryo) and EMD-2563 (ClpB wild type cryo). The corresponding FSC curves have also been deposited. The crystal structure atomic coordinates of *E. coli* ClpB E279A/E432A/E678A + ATP have been deposited in the PDB database (http://www.ebi.ac.uk/pdbe/) with entry code 4CIU. PDB models based on the EM maps have also been deposited with codes 4D2Q (BAP-E432A C6); 4D2U (BAP wild type C6) and 4D2X (BAP-Y503D C6).

## Acknowledgements

We thank Elena Orlova for help in image processing, Christos Savva for EM technical assistance, David Houldershaw for computing support and the Birkbeck EM group for useful discussions. This work was funded by Wellcome Trust grants 089050 and 079605 to H Saibil. E Kummer was supported by the Hartmut Hoffmann-Berling International Graduate School of Molecular and Cellular Biology. Y Oguchi was supported by a Humboldt fellowship.

## Additional information

### Funding

| Funder | Grant reference number | Author |
| --- | --- | --- |
| Wellcome Trust | 089050 | Helen R Saibil |
| Wellcome Trust | 079605 | Helen R Saibil |
| Hartmut Hoffmann-Berling International graduate school | | Eva Kummer |
| Humboldt fellowship | | Yuki Oguchi |

The funders had no role in study design, data collection and interpretation, or the decision to submit the work for publication.

### Author contributions

MC, EK, Conception and design, Acquisition of data, Analysis and interpretation of data, Drafting or revising the article; YO, IS, JK, Acquisition of data, Analysis and interpretation of data; PW, Acquisition of data, Contributed unpublished essential data or reagents; DKC, Conception and design, Analysis and interpretation of data, Contributed unpublished essential data or reagents; AM, BB, HRS, Conception and design, Analysis and interpretation of data, Drafting or revising the article

## Additional files

### Major datasets

The following datasets were generated:

| Author(s) | Year | Dataset title | Dataset ID and/or URL | Database, license, and accessibility information |
| --- | --- | --- | --- | --- |
| Carroni M, Kummer E, Oguchi Y, Wendler P, Clare DK, Sinning I, Kopp J, Mogk A, Bukau B and Saibil HR | 2014 | EM structure of BAP-E432A C6 | http://www.ebi.ac.uk/pdbe/entry/EMD-2555 | Publicly available at The Electron Microscopy Data Bank. |

| Carroni M, Kummer E, Oguchi Y, Wendler P, Clare DK, Sinning I, Kopp J, Mogk A, Bukau B and Saibil HR | 2014 | EM structure of BAP-E432A C1 | http://www.ebi.ac.uk/pdbe/entry/EMD-2556 | Publicly available at The Electron Microscopy Data Bank. |
|---|---|---|---|---|
| Carroni M, Kummer E, Oguchi Y, Wendler P, Clare DK, Sinning I, Kopp J, Mogk A, Bukau B and Saibil HR | 2014 | EM structure of BAP wild type C6 | http://www.ebi.ac.uk/pdbe/entry/EMD-2557 | Publicly available at The Electron Microscopy Data Bank. |
| Carroni M, Kummer E, Oguchi Y, Wendler P, Clare DK, Sinning I, Kopp J, Mogk A, Bukau B and Saibil HR | 2014 | EM structure of BAP wild type C1 | http://www.ebi.ac.uk/pdbe/entry/EMD-2558 | Publicly available at The Electron Microscopy Data Bank. |
| Carroni M, Kummer E, Oguchi Y, Wendler P, Clare DK, Sinning I, Kopp J, Mogk A, Bukau B and Saibil HR | 2014 | EM structure of BAP-Y503D C6 | http://www.ebi.ac.uk/pdbe/entry/EMD-2559 | Publicly available at The Electron Microscopy Data Bank. |
| Carroni M, Kummer E, Oguchi Y, Wendler P, Clare DK, Sinning I, Kopp J, Mogk A, Bukau B and Saibil HR | 2014 | EM structure of BAP-Y503D C1 | http://www.ebi.ac.uk/pdbe/entry/EMD-2560 | Publicly available at The Electron Microscopy Data Bank. |
| Carroni M, Kummer E, Oguchi Y, Wendler P, Clare DK, Sinning I, Kopp J, Mogk A, Bukau B and Saibil HR | 2014 | EM structure of HAP | http://www.ebi.ac.uk/pdbe/entry/EMD-2561 | Publicly available at The Electron Microscopy Data Bank. |
| Carroni M, Kummer E, Oguchi Y, Wendler P, Clare DK, Sinning I, Kopp J, Mogk A, Bukau B and Saibil HR | 2014 | EM structure of BAPtrap cryo | http://www.ebi.ac.uk/pdbe/entry/EMD-2562 | Publicly available at The Electron Microscopy Data Bank. |
| Carroni M, Kummer E, Oguchi Y, Wendler P, Clare DK, Sinning I, Kopp J, Mogk A, Bukau B and Saibil HR | 2014 | EM structure of ClpB wild type cryo | http://www.ebi.ac.uk/pdbe/entry/EMD-2563 | Publicly available at The Electron Microscopy Data Bank. |
| Carroni M, Kummer E, Oguchi Y, Wendler P, Clare DK, Sinning I, Kopp J, Mogk A, Bukau B and Saibil HR | 2014 | Crystal structure of *E. coli* ClpB E279A/E432A/E678A (SeMet) + ATP | http://www.ebi.ac.uk/pdbe-srv/view/entry/4ciu/summary | Publicly available at PDBe. |

The following previously published datasets were used:

| Author(s) | Year | Dataset title | Dataset ID and/or URL | Database, license, and accessibility information |
|---|---|---|---|---|
| Lee S, Sowa ME, Watanabe Y, Sigler PB, Chiu W, Yoshida M and Tsai FTF | 2003 | Crystal Structure Analysis of ClpBmophilus ClpB | http://www.rcsb.org/pdb/explore/explore.do?structureId=1QVR | Publicly available at PDB. |
| Wang F, Mei Z, Qi Y, Yan C, Hu Q, Wang J and Shi Y | 2011 | Structure of MecA121 and ClpC1-485 complex | http://www.rcsb.org/pdb/explore/explore.do?structureId=3pxg | Publicly available at PDB. |
| Jingzhi L and Bingdong S | 2001 | The Crystal Structure of ClpB N Terminal Domain, Implication to the Peptide Binding Function of ClpB | http://www.rcsb.org/pdb/explore/explore.do?structureId=1khy | Publicly available at PDB. |

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
