## [Decision Letter]

Thank you for sending your work entitled “Head-to-tail interactions of the coiled-coil domains regulate ClpB cooperation with Hsp70 in protein disaggregation” for consideration at *eLife*. Your article has been favorably evaluated by a Senior editor and 3 reviewers, one of whom, Andreas Martin served as a Guest Editor. All reviewers agreed that this is a beautiful paper that addresses an important question in the Clp/Hsp100 field and combines expertly performed EM with elegant biochemical experiments.

The following individuals responsible for the peer review of your submission have agreed to reveal their identity: Andy Martin (Guest Editor); Gabriel Lander (peer reviewer).

The Guest Editor and the other reviewers discussed their comments before we reached this decision, and the Guest Editor has assembled the following comments to help you prepare a revised submission.

AAA+ chaperones of the Clp/Hsp100 family play a major role in removing aggregated proteins in prokaryotic as well as eukaryotic cells, but their detailed mechanisms remain still largely unknown. The M-domain of ClpB has been revealed to have a crucial regulatory function through direct interactions with DnaK/Hsp70, but its location has never been clearly defined in an oligomeric assembly of ClpB or the related Hsp104.

In this manuscript, Saibil et al. used single particle EM to reconstruct the structure of BAP-ClpP with better definition than before, taking advantage of the side-view orientations of the complex. The image analysis was performed expertly, and the authors took care not to introduce model bias or over-refine their data. This allowed a clear distinction of the M-domain and consequently provided the first structural insight into the location, adopted orientations, and the dynamics of the M-domain in the ClpB hexamer. It is shown that the M-domain position at the periphery of the hexamer affects the availability of its motif 2 for interaction with DnaK/Hsp70. Based on asymmetric reconstructions, the authors found that the M-domains around the ring of the wild-type enzyme adopt differential orientations, ranging from a horizontal position with head-to-tail contacts between neighboring M-domains to a highly tilted position that allows Hsp70 binding. M-domain-dependent activation of the ClpB ATPase thus seems to occur in a coordinated wave-like fashion, and this manuscript presents an exciting new structural framework for the correlated regulation of ATP hydrolysis and DnaK-mediated substrate delivery to the ClpB disaggregase. Furthermore, the presented EM structural work is nicely complemented by elegant in-vitro biochemical experiments that analyze the spatial separation between motifs 1 and 2 for different M-domain conformations by means of FRET and characterize the roles of these motifs in regulating ATP hydrolysis and substrate processing.

In summary, this manuscript significantly furthers our mechanistic understanding of the M-domain's role in regulating the Hsp100 ATPase activity and the cooperation with Hsp70 for protein disaggregation, and it is well suited for publication in *eLife*.

None of the points below are major or would, if not addressed, jeopardize the acceptance of this manuscript for publication. Nevertheless, the authors should try to address them in order to make the paper even stronger than it already is.

1) In Figure 4, the authors present the measurements of FRET between an engineered Trp in motif 1 and a Cys-linked AEDANS label in motif 2 for wild-type and mutant ClpB. Overall, the data convincingly show that there is a difference in the spatial separation of motifs 1 and 2 for the repressed versus active conformation of MD. The only surprising observation for ClpB-WT is that the 50% increase in acceptor fluorescence upon oligomerization is not accompanied by a corresponding decrease in donor fluorescence. According to Figure 4—figure supplement 2, there is no change (increase) in the Trp donor fluorescence upon oligomerization, so it remains unclear why the acceptor fluorescence can double while the donor fluorescence stays unchanged. The authors should briefly discuss this point to strengthen the presented FRET data further.

2) In Figure 4 the authors present results of disulfide crosslinking between motifs 1 and 2 of neighboring MD domains. The hyperactive Y503D mutant with tilted MD also showed crosslinked multimers, albeit at lower concentrations, which the authors interpret as an indication for rapid MD fluctuations between the different conformations even in the hyperactive state. However, mutants with just a single Cys in either motif 1 or motif 2 also showed significant (presumably non-specific) crosslinking to form dimers (1st and 2nd lane). It is thus possible that the weaker crosslinking observed for the double Cys mutant of Y503D originates to a significant extent from such non-specific crosslinking rather than specific interactions of MDs that fluctuate between different conformations. The authors should therefore consider this background of non-specific crosslinking in their discussion.

3) Asymmetric reconstructions allowed the authors to reveal heterogeneity in the hexameric ring, with only 3 to 5 subunits in an ATP-binding competent conformation. This is a very exciting finding and consistent with studies on other AAA+ ATPases, showing that only a subset of the 6 subunits has nucleotide bound at any given time. To biochemically confirm their structural data, the authors analyzed nucleotide occupancy by measuring ADP binding in ITC experiments. However, it is unclear to me why they used ADP and not ATPγS, or ATP in combination with a hydrolysis-deficient Walker-B mutant of AAA-2. As the authors well know, ADP is not able to induce an active ring conformation that can bind substrate or the ClpP peptidase. The conformation of an ADP-bound subunit is likely more similar to an empty state than to an ATP-bound state. Therefore, the overall ring-conformation probed in the ITC experiment with ADP probably differs significantly from the hexamer state observed by EM in the presence of ATP, and it is questionable whether an ADP titration experiment indeed provides the right answer regarding the nucleotide occupancy. The authors should consider repeating the ITC experiment for instance with ATP and a Walker-B mutant AAA-2 that is deficient in hydrolysis but can adopt all possible conformations depending on its occupancy.

---

## [Author Response]

*1) In*
Figure 4*, the authors present the measurements of FRET between an engineered Trp in motif 1 and a Cys-linked AEDANS label in motif 2 for wild-type and mutant ClpB. Overall, the data convincingly show that there is a difference in the spatial separation of motifs 1 and 2 for the repressed versus active conformation of MD. The only surprising observation for ClpB-WT is that the 50% increase in acceptor fluorescence upon oligomerization is not accompanied by a corresponding decrease in donor fluorescence. According to*
Figure 4—figure supplement 2*, there is no change (increase) in the Trp donor fluorescence upon oligomerization, so it remains unclear why the acceptor fluorescence can double while the donor fluorescence stays unchanged. The authors should briefly discuss this point to strengthen the presented FRET data further*.

The reviewers are correct: we do not see a decrease in donor fluorescence while observing acceptor fluorescence when determining FRET upon ClpB oligomerization in absence of ATP. A correlation between loss and gain of donor and acceptor fluorescence is observed in presence of ATP and in all cases for the repressed ClpB-E432A variant, which shows stronger motif 1–motif 2 interactions. We cannot provide a straightforward explanation for this difference, which is now stated in the Results section.

*2) In*
Figure 4
*the authors present results of disulfide crosslinking between motifs 1 and 2 of neighboring MD domains. The hyperactive Y503D mutant with tilted MD also showed crosslinked multimers, albeit at lower concentrations, which the authors interpret as an indication for rapid MD fluctuations between the different conformations even in the hyperactive state. However, mutants with just a single Cys in either motif 1 or motif 2 also showed significant (presumably non-specific) crosslinking to form dimers (1st and 2nd lane). It is thus possible that the weaker crosslinking observed for the double Cys mutant of Y503D originates to a significant extent from such non-specific crosslinking rather than specific interactions of MDs that fluctuate between different conformations. The authors should therefore consider this background of non-specific crosslinking in their discussion*.

We agree with the reviewers that crosslink products observed for hyperactive ClpB-Y503D-K431C/S499C can in part be explained by non-specific background crosslinking and this information has been added to the manuscript. Higher crosslink products (trimers-hexamers) are, however, much more abundant than controls, supporting the original suggestion of rapid fluctuations in M-domain conformations, which can be captured by crosslinking.

*3) Asymmetric reconstructions allowed the authors to reveal heterogeneity in the hexameric ring, with only 3 to 5 subunits in an ATP-binding competent conformation. This is a very exciting finding and consistent with studies on other AAA+ ATPases, showing that only a subset of the 6 subunits has nucleotide bound at any given time. To biochemically confirm their structural data, the authors analyzed nucleotide occupancy by measuring ADP binding in ITC experiments. However, it is unclear to me why they used ADP and not ATPγS, or ATP in combination with a hydrolysis-deficient Walker-B mutant of AAA-2. As the authors well know, ADP is not able to induce an active ring conformation that can bind substrate or the ClpP peptidase. The conformation of an ADP-bound subunit is likely more similar to an empty state than to an ATP-bound state. Therefore, the overall ring-conformation probed in the ITC experiment with ADP probably differs significantly from the hexamer state observed by EM in the presence of ATP, and it is questionable whether an ADP titration experiment indeed provides the right answer regarding the nucleotide occupancy. The authors should consider repeating the ITC experiment for instance with ATP and a Walker-B mutant AAA-2 that is deficient in hydrolysis but can adopt all possible conformations depending on its occupancy*.

We agree that an ITC experiment using an ATPase-deficient ClpB Walker B mutant (ClpB-E279A/E678A) and ATP would have been preferable. We indeed performed such experiment but in repeated experiments the quality of the obtained ITC curves was not sufficient. We do not know the reason for this problem. We calculated the nucleotide occupancy, but these values are ambiguous because of the poor data quality and we prefer not to show these results. We therefore employed ADP, which provided much better results and allowed an accurate determination of the number of bound nucleotides.Author response image 1.Instrumental responses of successive injections of ATP (4.9 mM or 5.51 mM) into a solution of 380 μM or 496 μM ClpB-E279A/E678A. The molar ratio of ATP vs. ClpB-E279A/E678A hexamer is indicated. Raw isothermal titration calorimetry data are shown. Integrated data after base-line correction and fitting to the respective binding isotherms to a single binding model are given (lower panel). The number of ATP binding sites was calculated.

We think that nucleotide occupancy is likely to be similar between ADP and ATP states. Our calculated stoichiometries are in good agreement with numbers obtained for ClpX [Hersch, GL, et al, Cell, 2005], as each AAA domain of ClpB can bind four nucleotides.

We disagree that an ADP-bound subunit is more similar to an empty state than to an ATP-bound one. HX experiments of ClpB did not reveal differences in protection patterns in ADP and ATPγS, arguing that the conformations are similar [Oguchi, Y, et al, Nat Struct Mol Biol, 2012]. Importantly, both ADP and ATPγS binding led to substantial increase in HX protection compared to nucleotide-free ClpB oligomers, including Walker A motif regions. These data indicate that the overall conformational state of ADP-bound ClpB is more similar to the ATP-bound state than to the empty one.

It is true that the ADP conformation does not support substrate and ClpP interaction. Both interactions are mediated by flexible loop structures located either at the central channel or at the bottom of the ClpB ring. Nucleotide-driven conformational changes of these mobile elements do not however necessarily reflect substantial conformational changes within the bulk of the AAA+ domains. In fact, soaking of ClpX crystals with either ATPγS, ATP or ADP led to comparable structural changes and ADP, presumably generated by residual hydrolysis, fits best the electron density of an ATPγS-soaked crystal [Glynn, SE, et al, Cell, 2009]. Notably, soaking of ClpX with nucleotides led to asymmetric binding, supporting a model in which ADP also binds asymmetrically to ClpB hexamers. Similarly, binding of either ATP, ATPγS or ADP to ClpX led to the same degree of FRET changes in an experimental setup monitoring the formation of “L” subunits, which show tighter interaction between the large and small subunit of the AAA domain [Stinson, BM, et al, Cell, 2013]. Together these data indicate that nucleotides, irrespective of their identity, cause similar overall structural changes in AAA+ domains.